# Modulating Shrimp Tropomyosin-Mediated Allergy: Hypoallergen DNA Vaccines Induce Regulatory T Cells to Reduce Hypersensitivity in Mouse Model

**DOI:** 10.3390/ijms20184656

**Published:** 2019-09-19

**Authors:** Christine Y.Y. Wai, Nicki Y.H. Leung, Patrick S.C. Leung, Ka Hou Chu

**Affiliations:** 1School of Life Sciences, The Chinese University of Hong Kong, Shatin, Hong Kong SAR, China; 2Department of Paediatrics, School of Medicine, The Chinese University of Hong Kong, Shatin, Hong Kong SAR, China; 3Division of Rheumatology/Allergy, School of Medicine, University of California, Davis, CA 95616, USA

**Keywords:** vaccination, immunotherapy, tolerance induction, food allergy, hypoallergen, adoptive transfer

## Abstract

Shellfish allergy is one of the most common food allergies, with tropomyosin as the major cross-reactive allergen. However, no allergen-specific immunotherapy is clinically available. Recently, we designed two shrimp hypoallergens MEM49 and MED171. This study aimed to examine and compare the efficacy of the MEM49- and MED171-based DNA vaccines (pMEM49 and pMED171) in modulating shrimp allergy in a murine model of shrimp tropomyosin sensitivity. Intradermal immunization of BALB/c mice with pMEM49 or pMED171 effectively down-modulated allergic symptoms, tropomyosin-specific IgE levels, intestinal Th2 cytokines expression, and inflammatory cell infiltration. Both pMEM49 and pMED171 increased the frequency of regulatory T cells, but to a greater extent by pMED171 with upregulation of gut-homing molecules integrin-α4β7. The functionality of the pMED171-induced Treg cells was further illustrated by anti-CD25-mediated depletion of Treg cells and the adoptive transfer of CD4^+^CD25^+^Foxp3^+^Treg cells. Collectively, the data demonstrate that intradermal administration of pMED171 leads to the priming, activation, and migration of dermal dendritic cells which subsequently induce Treg cells, both locally and systemically, to downregulate the allergic responses to tropomyosin. This study is the first to demonstrate the potency of hypoallergen-encoding DNA vaccines as a therapeutic strategy for human shellfish allergy via the vigorous induction of functional Treg cells.

## 1. Introduction

Shellfish allergy is one of the most common food allergies, causing frequent anaphylaxis in both paediatric and adult populations [1,2]. The shrimp tropomyosin (Met e 1) has long been identified as the major cross-reactive shellfish allergen, with over 80% of shrimp-allergic patients sensitized to shrimp tropomyosin [3,4]. Current therapeutic recommendations for patients allergic to shellfish are limited to strict avoidance and epinephrine injection. Allergen-specific immunotherapy (AIT) is the only disease-modifying treatment that is currently available for respiratory allergies. However, AITs for IgE-mediated food allergies have significant efficacy limitations, associated side effects, debatable ability to achieve long-term protection following discontinuation of therapy, as well as adherence challenges [5,6,7,8,9,10]. No active immunotherapy for shellfish allergy is currently under investigation, even amidst the increasing prevalence of this type of food allergy. Approaches that mitigate the limitation of the existing AIT interventions and enhance permanent efficacy, therefore, prove necessary to be explored.

As an alternative, DNA vaccines could induce immune responses to the encoded antigens and are conceptually safer and more stable than conventional vaccines [11]. Recent advances in antigen designs and delivery methods have greatly enhanced the capacity and applications of DNA vaccines in infectious diseases, cancer, and allergies [12,13,14,15]. The potential clinical value of DNA vaccines for preventing and treating allergies is further highlighted by clinical trials that have demonstrated the safety and tolerability of DNA-based vaccines in patients with human infectious diseases and Japanese red cedar allergies [16,17,18]. Based on the efficacies of a multivalent peanut (Ara h1, h2, h3) LAMP DNA vaccine in murine model [19], a Phase 1 controlled trial is currently underway to further assess its safety and effectiveness in peanut-allergic adults (NCT02851277). This study will further highlight the potential of DNA vaccination in the clinical management of food allergies.

A recent study reported that AIT with native tropomyosin could lead to significant eosinophil infiltration in the gut and induce severe anaphylactic shock [20]. Thus, encoding unmodified tropomyosin in DNA vaccines may trigger allergic reactions. We reason that a combinatorial approach of hypoallergen and DNA-based formulation is a logical strategy for treating allergic diseases considering the non-sensitizing property of both hypoallergens and DNA plasmids. The extended in situ expression of hypoallergen by plasmid carrier also enables long-term allergen-specific immune memory with less frequent dosing and reduced dosages. We recently reported two designer hypoallergens, MEM49 and MED171, that were constructed by site-directed mutagenesis directed at the major IgE-binding epitopes of shrimp tropomyosin or by deleting these epitopes, respectively [21]. Herein, we evaluated the efficacy of MEM49-and MED171-encoding DNA vaccines (pMEM49 and pMED171) using an established murine model of shrimp tropomyosin-induced hypersensitivity [22,23] and demonstrated their capacity in the treatment of tropomyosin-induced shrimp allergy. We further demonstrated the potency of pMED171 in promoting regulatory T (Treg) cell responses and their functionality in limiting IgE-mediated allergic inflammation.

## 2. Results

### 2.1. pMEM49 and pMED171 Treatment Groups have Marked Reduction in Anaphylactic Symptoms and Intestinal Inflammation Upon Oral Shrimp Tropomyosin Challenge

BALB/c mice sensitized with shrimp tropomyosin using our routine intragastric sensitization and challenge protocol developed allergic statues with signs of anaphylaxis and diarrhea, as well as a rMet e 1-specific IgE level > 0.3 OD 450 nm after the first allergen challenge. After the second allergen challenge, anaphylaxis and diarrhea were much reduced in both pMEM49 and pMED171 treatment groups, whereas sham-treated mice (PBS- or naked pCI-Neo-treated) had signs of allergic symptoms and diarrhea similar to the degree observed after the first challenge (Table 1; Figure 1A,B). The median symptom scores after the first challenge were 3.5 for both PBS and vector control groups, while after the second challenge were 3.5 (PBS) and 3.0 (vector). This score was significantly decreased to 1.0 from 3.0 in the pMEM49-treated group, and to 0.0 from 3.5 in the pMED171-treated group (*p* < 0.001). The median diarrhea scores for the sham-treated groups were 1.0 (PBS) and 1.5 (vector) after the first challenge, compared to 2.0 for both groups after the second challenge. On the contrary, the diarrhea scores for pMEM49 and pMED171 groups were notably reduced to 1.0 after the second challenge compared to a score of 2.0 in the first challenge. However, the reduction of diarrhea score was statistically significant only in the pMED171-treated animals (*p* < 0.05), but not in the pMEM49 treatment group. 

pMEM49 and pMED171 therapy also markedly reduced the local inflammatory responses, as reflected by the number of infiltrations of both mast cells and eosinophils in the jejunum (Table 1; Figure 1C,D). Sham treated mice showed significant accumulation of mast cells at the crypt area (294 ± 46 and 239 ± 55 cells/mm^2^; *p* < 0.0001) and eosinophils along the villus (627 ± 157 and 618 ± 145 cells/mm^2^; *p* < 0.0001), compared to the naïve mice (62 ± 9 mast cells/mm^2^ and 222 ± 51 eosinophils/mm^2^). On the contrary, there were only 102 ± 22 and 111 ± 24 mast cells/mm^2^ in the jejunum of pMEM49- and pMED171-treated mice, respectively. Similarly, the number of eosinophils were 198 ± 92 and 296 ± 109 eosinophils/mm^2^ in the jejunum of pMEM49- and pMED171-treated mice, respectively. These values were significantly lower compared to the sham-treated mice (*p* < 0.0001) and statistically similar to the naïve mice. However, only the pMED171-treated mice had a significantly lower level of serological mMCP-1 when compared with the positive control groups (Figure 1E; *p* < 0.05), suggesting that pMED171 is more effective than pMEM49 in down regulating both the recruitment and activation of mast cells.

### 2.2. pMEM49 and pMED171 Therapy Reduces Shrimp Tropomyosin-Specific Serum IgE Level and Intestinal Th2-gene Expression

As expected, rMet e 1-specific IgE was not detected in the negative control mice. The levels of Met e 1-specific IgE among all challenged mice upon the first allergen challenge were similar (Figure 1F), which agreed with the occurrence of systemic anaphylactic symptoms in these groups (Figure 1A). After the second challenge, the rMet e 1-specific IgE levels of pMEM49- and pMED171-treated mice decreased significantly from 1.14 to 0.25 and 1.21 to 0.49 OD 450 nm, respectively (*p* < 0.01). On the contrary, the IgE levels remained high in the two positive control groups that only received sham treatment by PBS or naked plasmid. The IgE levels detected after the second challenge between the two positive control groups were statistically the same (*p* > 0.99). Importantly, IgE levels of the treatment groups after the second allergen challenge were statistically lower than the two positive control groups (*p* < 0.01).

All groups of mice were sacrificed the day after the second allergen challenge and the ileum sections of the small intestine were harvested for gene expression analysis. Spleens were also isolated for measuring cytokine levels upon rMet e 1 restimulation. A prominent repression in the expression of Th2-cytokines including *IL-4*, *IL-5,* and *IL-13*, and Th2 transcription factor *GATA-3* at mRNA level in the gut, as well as in Th2 cytokine production including IL-4 and IL-5 at protein level in the spleen was detected in both the pMEM49 and pMED171 treatment groups (*p* < 0.05). Conversely, significant upregulation of these Th2 cytokines and transcription factors (*p* < 0.01) and downregulation of the Th1 cytokines *IL-18* and *IFN-γ* were detected in the gut of PBS control group (Figure 1G and Figure 2A). Similarly, significantly higher production of IL-4 and IL-5, but decreased IFN-γ synthesis by splenocytes, was found in this group compared to the negative controls (Figure 1H; *p* < 0.05). In the vector control group, however, only intestinal *IL-13* expression but not *IL-4*, *IL-5,* or *GATA-3* was elevated when compared with naïve mice. *IL-18* expression was also elevated in this group of mice (*p* < 0.01), suggesting the induction of local Th1 responses by the empty vector. At the systemic level, production of IL-4, IL-5, and IFN-γ by the splenocytes obtained from the vector control group did not differ from the PBS control group. Interestingly, Th1 responses were not upregulated in pMEM49-treated mice whereas *IFN-*γ (but not *IL-18* or *IL-12*) (*p* < 0.05) was heightened at both mRNA and protein levels by pMED171. Collectively, these parameters and other inflammatory responses indicate a robust Th2 allergic status sustained in the sham-treated animals and the ability of pMEM49 and pMED171 in repressing Th2 responses.

### 2.3. pMED171 Markedly Enhances Intestinal Gut-Homing Treg Cell Responses

We also examined the Treg cell responses between different groups by flow cytometry, gene expression, and IHC analyses. In the spleen, we detected a significantly lower frequency of CD4^+^CD25^+^Foxp3^+^Treg cell in the positive control groups (Figure 2B,C; 5.5 ± 1.26% and 5.5 ± 1.41%) when compared with the negative controls (8.16 ± 1.23%; *p* < 0.001). Notably, Treg cell frequency was significantly higher in the pMEM49-treated mice when compared with the positive controls (9.78 ± 1.11%; *p* < 0.0001) but not compared to the negative controls (*p* = 0.104). Remarkably, the Treg cell population doubled in the pMED171-treated mice compared to the positive controls (14.0 ± 2.27%; *p* < 0.0001) and was also statistically higher than the negative control group (*p* < 0.0001). Similarly, both pMEM49 and pMED171 induced significantly more Foxp3^+^T cells (9.91 ± 1.10% and 12.9 ± 1.51%, respectively) compared to the PBS and vector controls (5.79 ± 0.92% and 6.42 ± 1.37%, respectively). These are concomitant with a higher IL-10 production in the spleen, as well as higher expression of *FOXP3* and *IL-10* in the ileum of pMED171 treatment group than the positive control groups (Figure 1H and Figure 2A; *p* < 0.01). To our surprise, IL-10 production was not elevated in the pMEM49 treatment group. Only *TGF-*β was found upregulated in the ileum of pMEM49 treatment group (*p* < 0.01), but not in the pMED171-treated mice. By IHC, accumulation of Foxp3^+^ cells in the Peyer’s patches at the ileum was evident in pMED171-treated mice, while they were either nondetectable or at very low numbers in the naïve, positive control, and pMEM49-treated mice (Figure 2D and data not shown).

Considering that DNA vaccine treatment was given intradermally, while Treg cell responses were detected also at the site of allergen exposure in the intestinal tissues, we investigated if the Treg cells were induced at the skin and subsequently migrated to the mesenteric lymph nodes (MLN) and small intestine, or if the vaccine-primed dendritic cells (DCs) trafficked to the MLN and induced gut-homing Treg cells at site. We therefore assessed the expression of different chemokine receptors and gut-tropic *integrin* α*4*β*7*. The expression of *CCR4*, *CCR7,* and *CCR8*, the chemokine receptors that express on peripheral Treg cells, was statistically similar among the treatment and control groups (Figure 2E), whereas *integrin* α*4*β*7* imprinted on Treg cells by the gut-associated DCs were significantly upregulated only in the pMED171 treatment group compared to the other control and pMEM49 treatment groups (*p* < 0.05). However, the expression of *CCR9*, another gut-homing chemokine receptor, remained low among all experimental groups.

### 2.4. Restoration of Intestinal Inflammatory Responses but not Th2 Cell Activity upon Treg Cell Depletion

We were intrigued by the specific induction of gut-homing Foxp3-expressing Treg cells by pMED171 treatment. It is also clear from our data that pMED171 treatment increased classical CD25^+^Foxp3^+^Treg cell population, expression, and activity to a much greater magnitude than pMEM49 treatment. As a first step to evaluate the role of pMED171-induced Treg cells in this intervention, we performed similar experiments in which Treg cells were depleted in PBS- and pMED171-treated groups using anti-CD25 monoclonal antibody. The antibody was given after the therapy and 24 h before the second allergen challenge. We first verified the depletion of Treg cells by flow cytometry, which showed a significantly lower number of CD4^+^CD25^+^Foxp3^+^Treg cells in the spleen (Figure 3A; *p* < 0.0001), and by qPCR, which verified the downregulation of *FOXP3* and *IL-10* in the small intestine of the antibody-injected groups (Figure 3B; *p* < 0.05). Histologically, such depletion of Treg cells led to the infiltration of mast cells (114 ± 3 mast cells/mm^2^) and eosinophils (602 ± 40 eosinophils/mm^2^), which were significantly higher compared to the pMED171-treated group without Treg depletion (51 ± 7 mast cells/mm^2^ and 299 ± 44 eosinophils/mm^2^; *p* < 0.0001) and statistically insignificant compared to the PBS-depleted and non-depleted groups (Figure 3C,D; *p* > 0.99). However, the depletion of Treg cells in the pMED171-treated mice neither restored rMet e 1-specific IgE level nor the expression of *IL-4* and *IL-13* (Figure 3E,F).

### 2.5. The Effect of pMED171 on Limiting Intestinal Inflammation Resides on CD4^+^CD25^+^Foxp3^+^Treg Cells

To further refine the role of the pMED171-induced Treg cells, CD4^+^CD25^+^ or CD4^+^CD25^-^T cells were isolated from the spleens of PBS-treated and pMED171-treated donor mice and transferred intravenously to sensitized mice 24 h before oral rMet e 1 challenge. The CD4^+^CD25^+^ cells from the donors for transfer were characterized by flow cytometry to confirm >85% purity of CD4^+^CD25^+^Foxp3^+^Treg cells and by ELISA for cytokine responses (i.e., undetectable IL-4, IL-13, IL-12, IFN-γ, and IL-10) upon rMet e 1 restimulation (data not shown). Mice transferred with CD4^+^CD25^+^Treg cells from either sham-treated or pMED171-treated donors showed upregulation of *CD25* in the gut when compared with mice without Treg transfer (Figure 4A; *p* < 0.01). A concomitant upregulation of *FOXP3* (Figure 5B; *p* < 0.001) and increased frequency of CD25^+^Foxp3^+^Treg cells in both the MLN (Figure 4C,D; 12.5 ± 1.5%; *p* < 0.001) and spleen (Figure 4E,F; 12.5 ± 2.3%; *p* < 0.0001) were detected only in mice receiving pMED171-Treg cells but not in the control groups, suggesting the preferential accumulation of pMED171-induced Treg cells at sites of allergen exposure and at the systemic level. In agreement with the results obtained from the Treg depletion experiment, transfer of Treg cells from pMED171-treated mice significantly suppressed the recruitment of mast cells and eosinophils to the site of inflammation (Figure 5A–D; *p* < 0.01). The number of mast cells (50 ± 25 mast cells/mm^2^) and eosinophils (249 ± 73 eosinophils/mm^2^) detected in the jejunum of mice receiving CD4^+^CD25^+^Foxp3^+^Treg cells from pMED171-treated donors were similar to that found in naïve mice (Table 1 and Figure 1). The transfer of pMED171 Treg cells also led to the significant downregulation of *IL-5*, *IL-13,* and *GATA-3* (Figure 5E; *p* < 0.05) in the small intestine, as well as decreased levels of rMet e 1-specific IgE (Figure 5F; *p* < 0.01) in the recipient mice compared to the control groups. Mice that received CD4^+^CD25^+^Foxp3^+^Treg cells from PBS-treated donor mice or CD4^+^CD25^−^T cells from pMED171-treated donor mice produced comparable serological levels of rMet e 1-specific IgE and expression of *Th2* genes as in mice that did not receive any T cell transfer. These suggest the specific role of pMED171-induced Treg cells in conferring the above suppressive effects.

## 3. Discussion

There is a growing body of evidence supporting the concept of AIT for food-allergic patients, but current interventions also have major challenges, such as significant efficacy limitations, associated adverse effects, debatable long-term protection, and poor adherence. Epicutaneous (EPIT) trials in peanut allergies usually reported insignificant clinical improvement despite a more favorable side effect profile [24,25]. Clinical trials directly comparing oral (OIT) and sublingual (SLIT) immunotherapy suggest the higher efficacy of OIT, but significantly more multisystem allergic reactions and medical intervention reported among OIT trials [26]. These approaches also only induced rapid, but temporary, Th2 suppression [27]. OIT with omalizumab allows more rapid and safe dose escalation, but significant adverse events have been reported following omalizumab cessation and a high drop-out rate remains [28,29,30]. Strategies of enhanced efficacy and safety are undeniably needed. In this study, we have obtained preclinical data on the efficacy of hypoallergen-encoding DNA vaccines in the clinical management of shellfish allergy in a mouse model of shrimp tropomyosin hypersensitivity. Furthermore, we also demonstrated the mechanism of action of hypoallergen-DNA vaccines with the induction of Treg cells in down-modulating shrimp allergy. The robust therapeutic capacities of both pMEM49 and pMED171 in attenuating the entrenched allergic responses is evident based on the following: (1) ability of pMEM49 and pMED171 to revert Th2 signaling; (2) the repression of IgE production and inflammatory cells activation in the small intestine; and (3) induction of Treg cells especially by pMED171. These immunological benefits are comparable to other OIT, SLIT, and EPIT methods using food extracts or food allergens in clinical trials and animal studies [5,31,32], as well as to DNA vaccines encoding the causal allergen [19,33]. No adverse reactions were observed throughout our treatment protocol, as reported in the Phase I clinical trial with a DNA vaccine against Japanese red cedar allergy [18]. These data collectively suggest hypoallergen DNA vaccine as a promising alternative clinical intervention in food allergy.

The most critical outcomes of AIT are to change the underlying pathology on local and systemic immune responses by inhibiting, deviating, and/or deleting Th2-mediated effector responses, as well as to offer persistent tolerance apart from only desensitization [34]. In this context, Treg cells involved in hampering the differentiation and effector functions of Th2 cells, inhibiting IgE synthesis, suppressing inflammatory cells, and modulating dendritic cells to an IL-10-producing tolerogenic phenotype represent a crucial marker in successful AIT [35]. The increase in functionally suppressive Foxp3^+^Treg cells by Der p 1 AIT was closely correlated to the reduction in total nasal symptom score [36]. Tolerance to peanut and egg is also associated with the number of CD25^+^Foxp3^+^Treg cells and IL-10-expressing CD4^+^T cells in children [37]. In contrast, a two-year grass pollen AIT did not lead to any increase in Treg cell responses and, thus, failed to offer long-term protection despite its effectiveness in reducing CRTH2^+^CCR4^+^CD27-CD4^+^Th2 cells and in desensitizing patients [38]. Similar to other successful AIT [39], the CD4^+^CD25^+^Foxp3^+^Treg cell population detected in the spleen was highly elevated in both pMEM49- and pMED171-treated mice when compared with the controls. Interestingly, the magnitude of systemic Foxp3^+^Treg cell expansion was much higher in the pMED171-treated mice. This result was accompanied by differential upregulation of *TGF-*β in the ileum of the pMEM49 treatment group, as opposed to *IL-10* and *FOXP3* in the pMED171-treated group, indicating the potential local induction of IL-10-independent LAG3^+^Foxp3^−^Treg cells and/or LAP^+^Th3 regulatory cells by pMEM49, and the classical CD25^+^Foxp3^+^Treg cells and/or IL-10^+^Tr1 cells by pMED171. Considering that the two vaccines were both constructed in pCI-Neo and delivered using the same modality, the intrinsic molecular differences might attribute to the preferential induction of different Treg cell subsets, which warrant further studies to precisely define their phenotypes and functionality. Moreover, our present experimental design, in which the treated animals were rechallenged after one week off treatment might not allow us to elucidate the durable effects of pMEM49 and pMED171, despite their clear contribution to enhanced Treg cells-mediated immune modulation. We are currently attempting to address this issue by evaluating the treatment effects of optimized pMEM49 and pMED171 at three months off treatment. We also consider that a combined administration of both pMEM49 and pMED171 may be an attractive approach, with respect to their ability to trigger a heterogeneous Treg response. Data from these experiments addressing improved and longer-term protection will be exciting.

The functional contributions of pMED171-induced Treg cells are further illustrated through the substantial restoration of inflammatory responses in the gut upon depletion of CD4^+^CD25^+^T cells despite pMED171 treatment. However, contrary to other studies, in which the depletion of CD4^+^CD25^+^T cells increased Th2 cytokine and IgE levels [40,41], depletion of Treg cells in pMED171-treated animals had no effects on IgE level or Th2 gene expression. This observation might be explained by the antagonizing effect exerted by Th1 cells, whereby we detected sevenfold upregulation of both *IL-18* and *IFN-*γ in pMED171-treated animals depleted of CD25^+^Treg cells (data not shown), and/or the preferential role of Treg cells in controlling allergic inflammation during its initiation, as reported in similar studies [42]. Indeed, data from our adoptive transfer experiment clearly indicate the suppressive capacities of the pMED171-induced CD4^+^C25^+^Treg cells, which include preventing the boost in IgE level, *Th2* genes expression, as well as hampering intestinal inflammation induced by tropomyosin challenge, even when sensitization is established. Such an effect could not be conferred by transferring CD4^+^CD25^+^T cells from sham-treated donors or CD4^+^CD25^-^non-Treg cells from pMED171-treated mice, highlighting that the regulatory capacity of these Treg cells requires the prior activation by pMED171 treatment and the specific inhibitory effects offered by these Treg cells. It is noteworthy that we also detected a significant increase of Treg cells in MLN and *FOXP3* upregulation in the ileum, despite the peripheral (i.e., intravenous) transfer of Treg cells, suggesting that the Tregs can migrate to the site of allergen exposure to protect against mast cell and eosinophil recruitment, as well as local Th2 responses. 

Apart from the *in vivo* suppressive functions of the induced Treg cells, the upregulation of only the gut-homing molecules, *integrin-α4β7*, but not the other chemokine receptors in the pMED171-treated animals suggests the capacity of this vaccine in priming antigen-presenting cells (e.g., dendritic cells and Langerhans cells) at the injection site (i.e., skin) that subsequently migrate to the draining lymph nodes, specifically MLN, to evoke T cell responses [43]. These T cells can then acquire both *integrin-*α*4*β*7* for intestinal imprinting and *FOXP3* expression in the MLN and upon trafficking of Foxp3^+^Treg cells into the small intestine, can confer tolerance [44]. It has been reported that the increased number of CD25^+^Foxp3^+^Treg cells in the nasal mucosa was closely associated to the clinical efficacy of Der p 1 AIT and suppression of seasonal allergic inflammation [36]. Together with our adoptive transfer data, this revitalizes the capacity of pMED171 in inducing and maintaining tolerogenic immune responses, initially from peripheral locales then to the site of allergen exposure, that drive the overall protection against shrimp allergy. Our data also support intradermal delivery as a preferred route due to the high frequency of cutaneous DC and skin DC as the only cell populations with migratory behavior to drive tolerance at the site of allergen exposure [45,46]. It is noteworthy that naked pCI-Neo “treatment” neither promoted Treg cell responses, repressed IgE synthesis, nor inflammatory responses, but only increased nonspecific Th1 responses. The promotion of regulatory responses thus requires the expression of MEM49 and MED171, and such responses can be translated beyond T-cell level to essentially offer clinically effective protection against shrimp allergy. Collectively, our study shows that vaccines encoding either MEM49 or MED171 have superior efficiency over naked DNA plasmid and comparable effects to other AIT modalities, and thus, provide a promising alternative strategy for future clinical applications.

We do realize that our current study has several limitations. Firstly, we could not detect significant changes in IgG_2a_ levels (data not shown) and *Th1 mRNA* or cytokine responses in both pMEM49 and pMED171 treatment groups, despite our previous report on their capacities in inducing inhibitory IgG antibodies [21]. On the contrary, we detected elevated Th1 responses in the vector controls with reduced Th2 cytokine responses compared to the PBS controls. Secondly, only pMED171 induced the classical CD25^+^Foxp3^+^Treg cells with upregulation in gut-homing signals, whereas neither the gut-homing signals nor migratory chemokine receptors were elevated in the pMEM49-treated animals. Different subsets of Treg cells, particularly in the small intestine, were also induced by pMEM49, but we have not defined their functionality as those induced by pMED171. Besides, we only demonstrated the effectiveness of pMEM49 and pMED171 in modulating tropomyosin sensitization without considering other minor shrimp allergens. Yet, heightening of Th1 cell activity promoted by the nonspecific effect of the plasmid CpG backbone has also been reported in a similar study [47] and this effect was only restricted at local mRNA level. Our data clearly illustrate that this nonspecific Th1 response could not be translated into repressing IgE synthesis and restricting inflammatory responses in the gut. We postulate that the Th1 responses might be inhibited by the regulatory signals triggered by pMEM49 and pMED171. Further, the heterogenicity in the regulatory pathways between pMEM49 and pMED171 treatment can be attributed to the intrinsic molecular differences between the two hypoallergenic molecules. Future mechanistic studies detailing these hypotheses, and experiments isolating and characterizing the specific LAG^+^ and/or LAP^+^ regulatory T cells induced by pMEM49, would yield more profound understanding of the immunomodulatory mechanisms involved. Moreover, most studies have clearly illustrated that sensitization rates to other minor shrimp allergens range only between 10–15% and are often independent of tropomyosin sensitization [48]. It is also important to note that 94% of the IgE against whole shrimp extract is directed to tropomyosin and IgE sensitization to tropomyosin is strongly associated with a positive oral food challenge to shrimp [49,50,51]. Taken together with the fact that tropomyosin contributes to shellfish cross-reactivity [52], we believe that this study provides proof-of-concept data that establish tropomyosin hypoallergen DNA vaccines as an applicable option for treating most shellfish-allergic subjects.

This is the first study demonstrating the immunotherapeutic feasibility of hypoallergen DNA-based vaccine formulations against shrimp tropomyosin-induced hypersensitivity, and the capacity of this formulation in inducing functional Treg cells to repress Th2 cell activities, IgE synthesis, and limit inflammatory cell responses in the gut are transferrable to other animals. Although the use of DNA vaccines in food allergy is still in its infancy and the more precise modulatory mechanism of such therapeutic intervention remains to be fully elucidated, we believe these issues can be easily addressed by cell-specific mechanistic studies and human clinical trials. We anticipate that hypoallergen- and DNA-based immunotherapy will be a promising AIT approach in the clinical management of food allergy and other allergic disorders at large.

## 4. Materials and Methods

### 4.1. Mice

BALB/c mice acquired from The Chinese University of Hong Kong were maintained in pathogen-free conditions and on a shrimp-free diet at the Laboratory Animal Services Centre. The animal protocols were approved by the Animal Experimentation Ethics Committee of The Chinese University of Hong Kong (14/035/MIS, approved on 9 April 2014) and conducted under licenses granted from the Department of Health, Hong Kong Special Administrative Region, China.

### 4.2. DNA Vaccine Preparation

Nucleotide sequences encoding MEM49 and MED171 were cloned into pCI-Neo mammalian expression vector (Promega, Medison, WI, USA) via the *XhoI/XbaI* and *EcoRI/XhoI* restriction sites, respectively, and the reading frame was validated by DNA sequencing. The resulting DNA vaccines, pMEM49 and pMED171, were transformed into *Escherichia coli* DH5α, propagated, and purified using NucleoBond PC 500 (Macherey-Nagel, Düren, Germany).

### 4.3. Study Design

The first experiment was designed to illustrate the therapeutic efficacies of pMEM49 and pMED171 (Figure 6A). BALB/c mice (*n* = 8) in the PBS control, vector control, and treatment groups were intragastrically sensitized with 0.1 mg recombinant Met e 1 (rMete 1; prepared as described previously) and 10 μg cholera toxin (Sigma-Aldrich, St. Louis, MO, USA) on days 0, 12, 19, and 26. On day 33, all mice received the first rMet e 1 intragastric challenge (0.5 mg). Mice in the treatment groups then received three intradermal treatments of 100 μg pMEM49 or pMED171 on days 40, 52, and 59, while those in the PBS and vector control groups received equal volume of PBS and equal dose of empty pCI-Neo, respectively. One week after the last treatment (day 66), all mice received the second 0.5 mg rMet e 1 intragastric challenge. Mice in the negative control group were given PBS throughout the experiment. Mice were sacrificed for organ recovery one day after the second allergen challenge.

In the second experiment, Treg cells were depleted by means of anti-mouse CD25 antibody injection (clone PC61.5, eBioscience, Santa Calra, CA, USA) to study the therapeutic contributions of the pMED171 therapy-induced Treg cells (Figure 6B). A negative control group remained non-sensitized and untreated throughout this experiment. Following intragastric sensitization and challenge as described above, BALB/c mice were allocated into four different experimental groups (*n* = 8 per group): (1) the PBS control group receiving placebo treatment; (2) the PBS + anti-CD25 group receiving placebo treatment and intraperitoneal injection of 100 μg anti-mouse CD25 antibody on day 65 (i.e., 24 h before the second rMet e 1 oral challenge); (3) the pMED171 treatment group receiving three 100 μg pMED171 intradermal treatments; and (4) the pMED171 + anti-CD25 group receiving pMED171 treatment and anti-mouse CD25 antibody injection on day 65. One week after the last treatment (day 66), all mice in the experimental groups were orally challenged with 0.5 mg rMet e 1 and all mice were sacrificed on the next day for assessment.

The third experiment was designed to characterize the specific mechanistic roles of the pMED171 therapy-induced Treg cells (Figure 6C). BALB/c donor mice were sensitized and treated using the sample protocol as described in experiment 1 (PBS-treated donor mice *n* = 12; pMED171-treated donor mice *n* = 12). The CD4^+^CD25^+^Treg cells were sorted from spleen cells in each group using the Dynabeads FlowComp Mouse CD4^+^CD25^+^Treg cells Kit (Life Technologies AS, Norway) according to the manufacturer’s instructions. Purity of the sorted Treg cells defined as CD4^+^CD25^+^Foxp3^+^cells was >85%, as checked by flow cytometry. In total, 5 × 10^6^ Treg cells suspended in 100 μL PBS were then intravenously transferred to rMet e 1-sensitized nontreated mice (recipient mice *n* = 6 per group) at the tail vein 24 h before intragastric rMet e 1 challenge, which was given on day 33. Mice that were sensitized and challenged to rMet e 1 but not adoptively transferred with any T cells (no transfer group) or transferred with CD4^+^CD25^−^T cells isolated from pMED171-treated donor mice (CD4^+^CD25^-^control group) were included in this experiment. All mice were sacrificed on day 34 for assessment.

### 4.4. Assessment of Hypersensitivity Responses and Condition of Feces

Anaphylactic reactions were scored in a blinded manner 30 to 40 min post-challenge as described: 0, no symptoms; 1, scratching and rubbing around snout and head; 2, puffiness around eyes and snout, reduced activity, and/or decreased activity with increased respiratory rate; 3, wheezing, labored respiration, or cyanosis around the mouth and tail; 4, no activity after prodding or convulsion; and 5, death. Condition of feces was also assessed for 3 h after challenge and scored: 0, hard feces; 1, soft feces; 2 liquid feces; and 3, white, mucus like feces.

### 4.5. Measurement of Antibody, mMCP-1, and Cytokine Levels

Blood collected 24 h after allergen challenge was used for measuring the levels of rMet e 1-specific IgE, IgG_1_, and IgG_2a_ at dilutions of 1:10, 1:10,000, and 1:200, respectively, on 96-well ELISA Nunc MaxiSorp flat-bottom plates (Invitrogen, Carlsbad, CA, USA), as described previously. Mouse mast cell protease-1 (mMCP-1) level was determined using sandwich ELISA kits (eBioscience, Santa Clara, CA, USA) as per the manufacturer’s instructions. Single cell suspension from individual spleens were prepared as previously described [53], and restimulated with 0.05 mg rMet e 1 at a density of 5 × 10^6^ cells/mL for 72 h at 37 °C in a 5% CO_2_ incubator. Culture supernatants were then collected for the measurement of cytokine levels using sandwich ELISA kits (BD Biosciences, San Diego, CA, USA) in accordance with the manufacturer’s instructions.

### 4.6. Measurement of Regulatory T Cells

To measure Treg cells, spleens and mesenteric lymph nodes (MLNs) were teased into single-cell suspension and 1.0 × 10^6^ cells were blocked with anti-CD16/32. Surface staining was performed with CD4-FITC (clone RM4-5) and CD25-APC (clone PC61.5), followed by intracellular Foxp3 staining using the fixation and permeabilization buffers, and Foxp3-PE antibody (clone FJK-16s). All buffers and antibodies were purchased from eBioscience. Flow cytometry was performed on BD FACSVerse (BD Biosciences) and analyzed using the FACSuite software. Cells were gated on lymphocytes using FCS/SSC and at least 1 × 10^5^ events were collected for each sample. Frequency of Treg cells in the spleen and MLN was given as a percentage of CD25^hi^Foxp3^hi^ cells within the CD4^hi^ population.

### 4.7. Intestinal Gene Expression

Ileum sections of the small intestine were stored in RNAlater RNA stabilization reagent (QIAGEN, Valencia, CA, USA) at −20°C until use. Total RNA was extracted using the RNeasy mini kit (QIAGEN) and reverse-transcribed into cDNA using m-MuLV reverse transcriptase (Genesys Daly City, CA, USA). RT-PCR was performed on the ABI Step One Plus Real-Time PCR System (Applied Biosystems, Waltham, MA, USA) using specific primers, 100 ng cDNA, and Fast SYBR green PCR master mix (Life Technologies, Carlsbad, CA, USA). The relative levels of mRNA expression were determined with reference to HPRT-1 expression in each sample and naïve controls using the comparative *C*_t_ method.

### 4.8. Immunohistochemistry

Jejunum and ileum sections were fixed in 4% paraformaldehyde and embedded in paraffin as described [22]. Mast cells in the jejunum were visualized using the Naphthol AS-D chloroacetate esterase staining kit (Sigma-Aldrich). Eosinophils in the jejunum and Foxp3^+^T cells in the ileum were identified by immunohistochemistry (IHC) by incubating tissue sections with anti-mouse major basic protein monoclonal antibody (1:1500; provided by Jamie J. Lee, Mayo Clinic, Scottsdale, AZ, USA) and anti-mouse/rat Foxp3 monoclonal antibody (clone FJK-16s; 1:100; eBioscience), respectively. The sections were then developed with HRP conjugated goat anti-rat IgG antibody (Millipore, Billerica, MA, USA), DAB color-developing solution (Dako, Glostrup Municipality, Denmark), and counter-stained with hematoxylin gill number 3 (Sigma-Aldrich). For all analyses, two tissue sections were stained. Quantification of mast cells and eosinophils was performed by analyzing three random sections per stained sample on the ImageJ software and expressed as number of cells per mm^2^ of mucosa.

### 4.9. Statistical Analysis

Results are expressed as mean ± standard deviation (SD) of two independent experiments. GraphPad Prism 6 (GraphPad Software Inc., La Jolla, CA, USA) was used for statistical analyses and graphical presentation. The data were assessed for normal distribution and differences among different experimental groups were analyzed by one-way ANOVA, followed by Bonferroni post-test. For data that were not normally distributed, statistical differences between groups were assessed by the nonparametric Mann–Whitney U test. Each P value was adjusted to account for multiple comparisons and *p* values of less than 0.05 were considered significant, and are illustrated with asterisks or different alphabets in the figures and tables.

## Figures and Tables

**Figure 1 ijms-20-04656-f001:**
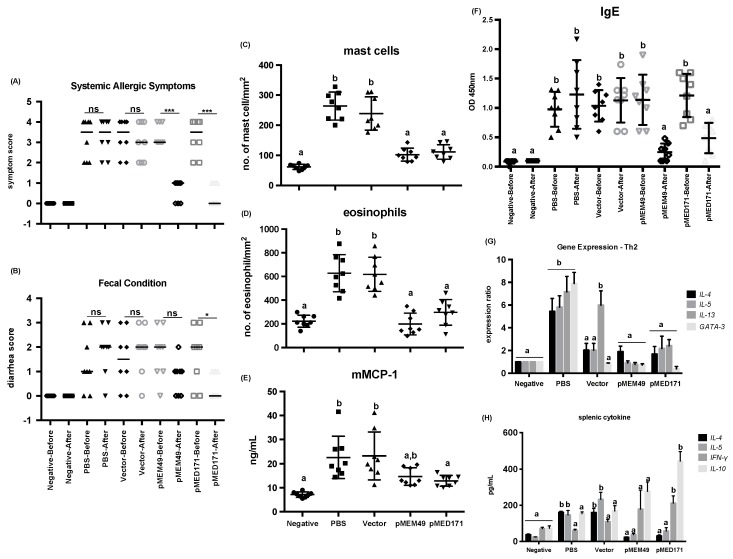
Therapeutic efficacies of pMEM49 and pMED171. (**A**) Scores of systemic allergic responses and (**B**) fecal condition recorded within 40 min post-challenge. Data are presented as individual data points denoted by different symbols for each experimental group with median and between group differences were assessed by Mann–Whitney U test. * *p* < 0.05; *** *p* < 0.001 and ns = not significant. Quantification of (**C**) mast cells per mm^2^ of crypt area and (**D**) eosinophils per mm^2^ of villus in the small intestine. Levels of serological (**E**) mouse mast cell protease-1 and (**F**) tropomyosin-specific IgE measured by ELISA. (**G**) Expression of Th2 cytokines and transcription factors in the small intestine detected by qPCR and (**H**) levels of cytokines produced by spleen cells restimulated by rMet e 1, measured by sandwich ELISA. Note the significant reductions in Th2 systemic and local responses upon pMEM49 and pMED171 treatment. Data are shown as individual data points denoted by different symbols for each experimental group with mean ± SD. Statistical differences among experimental groups were determined by Bonferroni post-test after one-way ANOVA; groups denoted by the same alphabet are not significantly different, while those denoted by different alphabets are statistically different (*p* < 0.05).

**Figure 2 ijms-20-04656-f002:**
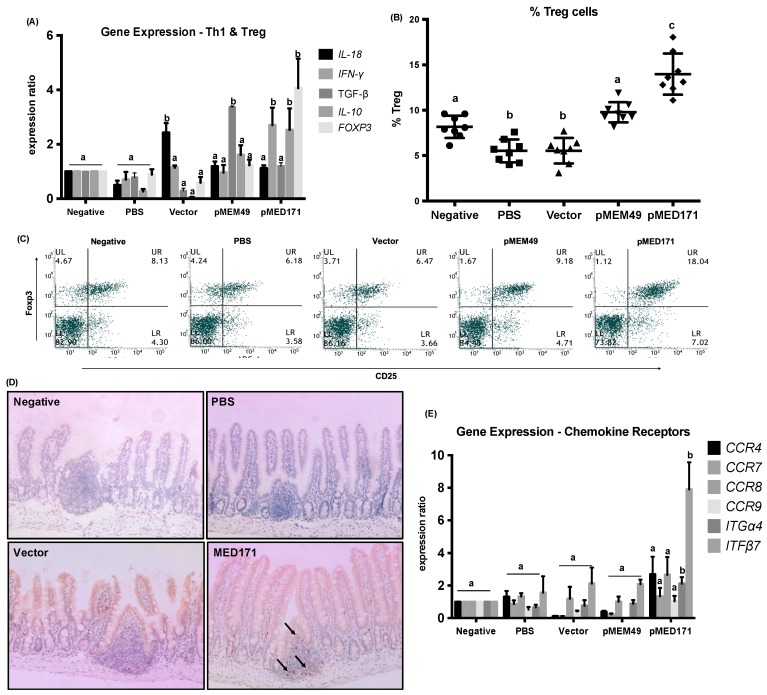
Th1, Treg, and chemokine responses with pMEM49 and pMED171 treatment. (**A**) Local expression of Th1 and Treg cytokines and transcription factors in the small intestine detected by qPCR. (**B**) Frequency of splenic Treg cells gated as CD4^hi^ and CD25^+^Foxp3^+^cells. Individual data points denoted by different symbols for each experimental group are shown. (**C**) Representative flow plot showing the frequency of splenic Treg cells in control and treatment groups. (**D**) Representative photomicrographs of Foxp3^+^T cells in the ileum (arrows indicate Foxp3^+^T cells stained brown in IHC; magnification = 200×). (**E**) Local expression of chemokine receptors and gut-homing signals in the small intestine detected by qPCR. Note the significant enhancement of Treg activities with pMEM49 and pMED171 treatment, despite the differential magnitude and type of Treg responses. Data are shown as mean ± SD. Statistical differences among different experimental groups were determined by Bonferroni post-test after one-way ANOVA; groups denoted by the same alphabet are not significantly different, while those denoted by different alphabets are statistically different (*p* < 0.05).

**Figure 3 ijms-20-04656-f003:**
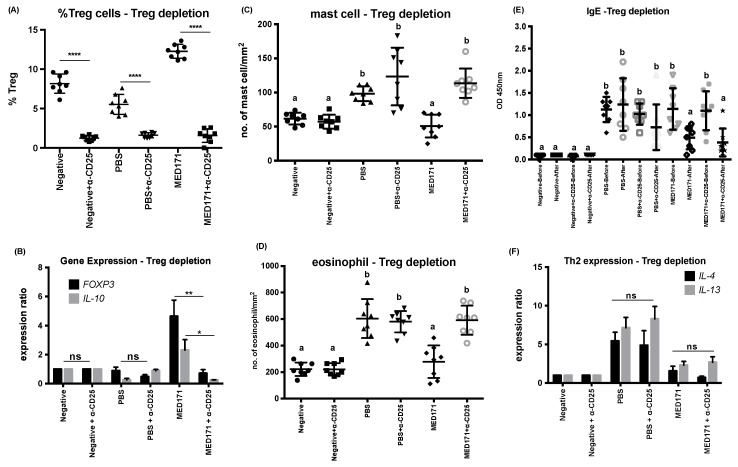
Immunological responses with Treg cell depletion. Intraperitoneal injection of anti-CD25 antibody 24 h prior to allergen challenge led to (**A**) significant reduction of splenic Treg cell frequencies below 2% and (**B**) significant downregulation of both *FOXP3* and *IL-10* in the gut. Statistical differences between Treg-depleted and non-depleted groups were assessed by Mann–Whitney U test. * *p* < 0.05, ** *p* < 0.01, and **** *p* < 0.0001. Quantification of (**C**) mast cells per mm^2^ of crypt area and (**D**) eosinophils per mm^2^ of villus in the small intestine. Levels of serological (**E**) tropomyosin-specific IgE measured by ELISA and (**F**) local expression of Th2 cytokines in the small intestine detected by qPCR. Note that Treg cell depletion in pMED171-treated animals only led to restoration of local inflammatory responses but not systemic Th2 activities. Data are shown as individual data points denoted by different symbols for each experimental group with mean ± SD. Statistical differences among different experimental groups were determined by Bonferroni post-test after one-way ANOVA. Groups denoted by the same alphabet are not significantly different, while those denoted by different alphabets are statistically different (*p* < 0.05). ns = not significant.

**Figure 4 ijms-20-04656-f004:**
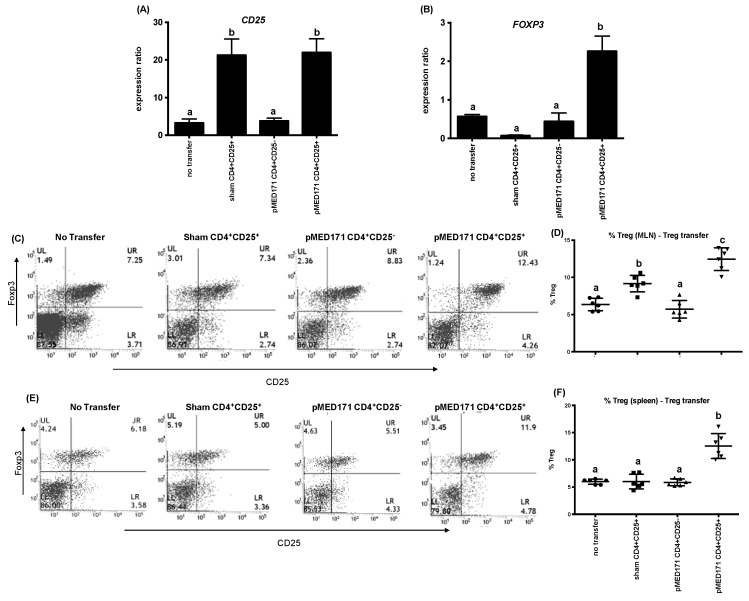
Treg cell responses in the recipient mice in T cell adoptive transfer experiment. Intravenous transfer of CD4^+^CD25^+^Treg cells led to intestinal (**A**) nonspecific upregulation of CD25 and (**B**) specific FOXP3 upregulation only in mice receiving Treg cells from a pMED171-treated donor (**C**) Representative flow plot analysis and (**D**) percentage of CD25^+^Foxp3^+^ Treg cells gated within the CD4^+^ in the mesenteric lymph nodes (MLN). (**E**) Representative flow plot analysis and (**F**) percentage of CD25^+^Foxp3^+^Treg cells gated within the CD4^+^ in the spleen. Note that the peripheral transfer of Treg cells can lead to both systemic and local migration of these cells to the site of allergen exposure to offer protection. Data are shown as individual data points denoted by different symbols for each experimental group with mean ± SD. Statistical differences among different experiment groups were determined by Bonferroni post-test after one-way ANOVA; groups denoted by the same alphabet are not significantly different, while those denoted by different alphabets are statistically different (*p* < 0.05).

**Figure 5 ijms-20-04656-f005:**
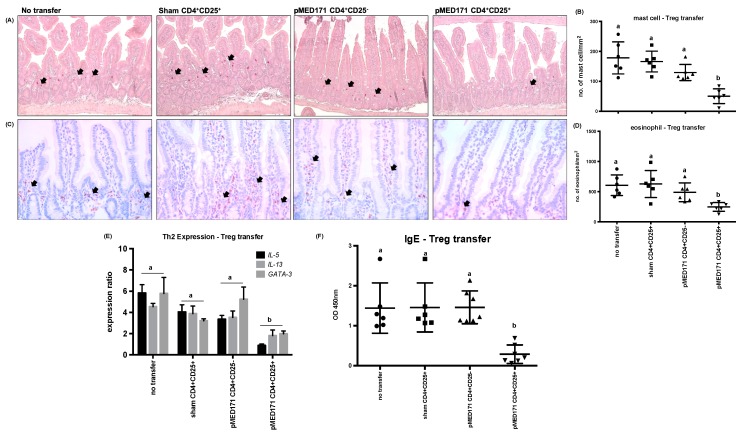
Immunological responses with Treg cell transfer. (**A**) Representative photomicrographs of mast cells (arrows indicate mast cells stained red in chloroacetate esterase staining; magnification = 200X). (**B**) Quantification of mast cells per mm^2^ of crypt area. (**C**) Representative photomicrographs of anti-MBP-stained eosinophils (arrows indicate eosinophils stained brown in IHC; magnification = 400X). (**D**) Quantification of eosinophils per mm^2^ of villus. (**E**) Local expression of Th2 cytokines and transcription factors in the small intestine detected by qPCR and (**F**) levels of serological tropomyosin-specific IgE measured by ELISA. Note that only the transfer of CD4^+^CD25^+^Treg cells from pMED171-treated donors led to significant reduction in intestinal inflammation, Th2 responses, and allergen-specific IgE titer. Data are shown as individual data points denoted by different symbols for each experimental group with mean ± SD. Statistical differences among different experimental groups were determined by Bonferroni post-test after one-way ANOVA; groups denoted by the same alphabet are not significantly different, while those denoted by different alphabets are statistically different (*p* < 0.05).

**Figure 6 ijms-20-04656-f006:**
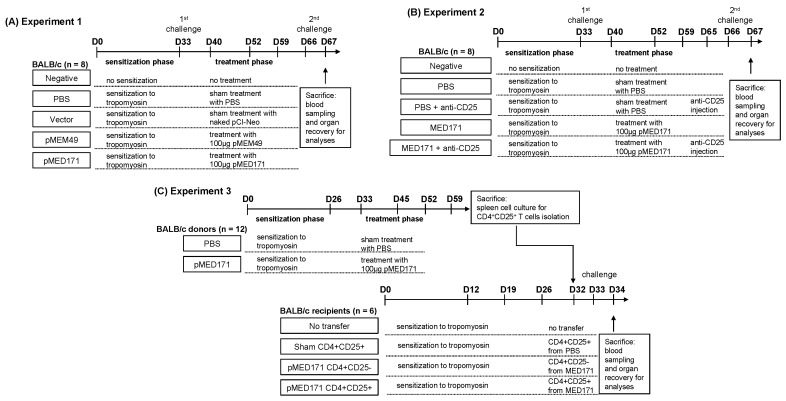
Study design of animal experiments. (**A**) Experiment 1 for assessing the therapeutic efficacies of pMEM49 and pMED171. BALB/c mice (*n* = 8) were randomly divided into control and treatment groups. Subsequent to the sensitization phase and first challenge, animals were either sham-treated with PBS, naked vector, pMEM49, or pMED171. Second tropomyosin challenge was delivered one week after the last treatment. (**B**) Experiment 2 for assessing the function of pMED171-induced Treg cells by anti-CD25 antibody-mediated Treg cell depletion. Following sensitization and treatment phases, selected groups of animals (*n* = 8 per group) were dosed intraperitoneally with anti-CD25 antibody 24 h before the second tropomyosin challenge. (**C**) Experiment 3 for assessing the functional roles of pMED171-induced Treg cells by adoptive transfer method. CD4^+^CD25^−^T cells and CD4^+^CD25^+^Treg cells were isolated from donor mice receiving tropomyosin sensitization with/without pMED171 treatment (*n* = 12). Cells were transferred intravenously to recipient mice sensitized with tropomyosin 24 h before tropomyosin challenge (*n* = 6 per group). Animals from all experiments were sacrificed 24 h after allergen challenge for blood sampling and organ recovery for analyses.

**Table 1 ijms-20-04656-t001:** Systemic Th2 inflammatory responses in control and treatment groups. Symptom and diarrhea scores are presented as median while other data are shown as mean ± SD. Statistical differences of the symptom and diarrhea scores before and after treatment within each experimental group were assessed by Mann–Whitney U test. * *p* < 0.05 and *** *p* < 0.001. Differences of other parameters among different experimental groups were determined by Bonferroni post-test after one-way ANOVA; groups denoted by the same alphabet are not significantly different, while those denoted by different alphabets are statistically different (*p* < 0.05).

Group	Treatment	Symptom Score	Diarrhea Score	Mast Cell Count (cells/mm^2^)	Eosinophil Count (cells/mm^2^)	mMCP-1 (ng/mL)	rMet e 1-Specific IgE (OD 450 nm)	Splenic IL-4 (pg/mL)	Splenic IL-5 (pg/mL)
**Negative**	**Before**	0.0	0.0	62 ± 9 ^a^	221 ± 51 ^a^	7.1 ± 1.1 ^a^	0.09 ± 0.04 ^a^	38 ± 4 ^a^	25 ± 1 ^a^
	**After**	0.0	0.0				0.10 ± 0.00 ^a^		
**PBS**	**Before**	3.5	1.0	294 ± 46 ^b^	627 ± 157 ^b^	22.6 ± 8.8 ^b^	0.98 ± 0.30 ^b^	161 ± 6 ^b^	147 ± 25 ^b^
	**After**	3.5	2.0				1.23 ± 0.58 ^b^		
**Vector**	**Before**	3.5	1.5	239 ± 55 ^b^	618 ± 145 ^b^	23.2 ± 9.9 ^b^	1.04 ± 0.27 ^b^	160 ± 22 ^b^	233 ± 39 ^b^
	**After**	3.0	2.0				1.13 ± 0.38 ^b^		
**pMEM49**	**Before**	3.0	2.0	102 ± 22 ^a^	198 ± 92 ^a^	14.6 ± 3.6 ^a,b^	1.14 ± 0.43 ^b^	24 ± 2 ^a^	38 ± 7 ^a^
	**After**	1.0 ***	1.0				0.25 ± 0.15 ^a^		
**pMED171**	**Before**	3.5	2.0	111 ± 24 ^a^	296 ± 109 ^a^	12.9 ± 2.2 ^a^	1.21 ± 0.37 ^b^	31 ± 5 ^a^	58 ± 18 ^a^
	**After**	0.0 ***	1.0 *				0.49 ± 0.26 ^a^		

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
