# Peer review of "Modulating Shrimp Tropomyosin-Mediated Allergy: Hypoallergen DNA Vaccines Induce Regulatory T Cells to Reduce Hypersensitivity in Mouse Model"

_ijms, 2019, doi:10.3390/ijms20184656_

Round 1

Reviewer 1 Report

In this work authors have shown the effect of two DNA vaccines (pMEM49 and pMED171) on the reduction of anaphylactic symptoms and intestinal inflammation in a mouse model of oral shrimp allergy. In addition, they have nicely demonstrated a marked effect of the pMED171 vaccine in the specific induction of CD25+Foxp3+ Tregs. However, I have some concerns regarding this manuscript that should be clarify before its publication.

The statistical analysis included in this manuscript is really difficult to follow. In table 1, statistics has not been included, although if it has been referred in the text. Could you please include the statistical analysis in this table? In the same table, the level of significance is not taken in account and some data are expressed with two decimals, while others only have one or do not have any. Could you be consistent in your level of significance? In Figure 2 A and F, differences are exclusively shown between “before” and “after” treatment. Could you specify it in the footnote? However, in the rest of the figures I am not able to understand what the asterisks refer to. Do they show differences between groups? Between treated groups and vector group? Or PBS group? Could you clarify in the footnote the comparisons showed and what asterisks represent, please? Similar observation should be taken in account for the rest of the figures. Figure 2B. There are not significant differences between treated groups before and after treatments? Actually, it looks like there are differences. The footnote of the figure 2 reads “data are shown as mean ± SD”. Is that also true for scores (Fig 2 A and B)? I believe that only medians are shown. Regarding the cytokine quantification, the experimental origin of these samples should be clarified. Line 383 reads “Mice were sacrificed for organ recovery one day after the second allergen challenge” However, experimental details regarding splenocyte culture and whether cytokines are measure in the supernatant is not described. Are the splenocytes ex vivo re-stimulated with Met e 1? Could you please clarify it? Regarding IgE quantification. When the blood samples were collected? Before or after challenge? From line 156 to line 161 authors point out that pMED171 treatment increases the total number of CD25+Foxp3+ Tregs compared with the negative control as well as an increased expression of IL10 and Foxp3. On the other hand, pMEM49 also increased CD25+Foxp3+ T regs and expression of TGF-beta compared to PBS and vector groups. Taken together, it looks like different subsets of Tregs are induced by each treatment. Have you considered that pMED171 could increase both conventional Foxp3+ Treg and CD25+ IL10-producings Tregs (Treg type Tr1), while pMEM49 administration could be able to induce LAP+ Foxp3+ Tregs (Treg type Th3)? Do you have any data consistent with this idea? I believe that the study of the surface expression of LAP on these cells could be very useful in the determination of the different Treg subsets generated by the two different treatments. Could you provide some data regarding that? Could the combined administration of both pMEM49 and pMED171 be a better strategy for shrimp allergy treatment? Regarding CD25+ Tregs used in the transfer experiment, have you characterized them? Authors show that these cells are FoxP3+. Are they also IL10+? Is it possible that you are transferring two T reg subsets with different immunosuppressive effect (Foxp3+ and Tr1)? Could that affect the results? Finally, authors have shown how Tregs induced by PMED171 treatment expressed gut homing markers. Could you provide any result characterizing the phenotypical profile of the dendritic cells in the draining lymph nodes after intradermal administration of the vaccine? Are these cells PDL2+? In order to fully understand the generation Tregs and the induction of homing markers, a characterization of the dendritic cells that primed these cells could be really useful and this manuscript could increase its impact.

Author Response

Response 1: We thank the reviewer for the comment. Table 1 has been revised to include all statistical analyses. Statistical differences assessed by the non-parametric Mann-Whitney U test are illustrated by asterisks while those measured by one-way ANOVA followed by Bonferroni posttest were illustrated by alphabets. Groups denoted by the same alphabets are not significantly different while groups denoted by different alphabets (e.g. a or b) are significantly different (p < 0.05). The significant figures after decimals are now consistent for each of the columns of data.

Response 2: We thank the reviewer for the comment. Considering that alphabet annotation is an alternative and standard illustration for one-way ANOVA and multiple range test, we have revised Figures 2C-H, 3,4C-F, 5 and 6, and used alphabets to replace asterisks to better illustrate the statistical differences among the experimental groups. Comparisons were made among experimental groups using one-way ANOVA followed by Bonferroni posttest. Groups denoted by the same alphabets are not significantly different while groups denoted by different alphabets (e.g. a, b or c) are significantly different (p < 0.05). These are also described in the figure legends of the revised manuscript.

Response 3: The differences in diarrhea score among all experimental groups were not significantly different when assessed by one-way ANOVA. We have re-analysed the data using the non-parametric Mann-Whitney U test. The reduction of diarrhea score was statistically significant only in the pMED171-treated animals (P < 0.05) but not in the pMEM49 treatment group. Figure 2B, Table 1, lines 79-87 and lines 506-509 within text have been revised accordingly.

Response 4: We thank the reviewer for the comment. We have revised the two figures. The symptom and diarrhea scores in figures 2A and 2B are now presented as median and are also clarified in the figure legend. Similar changes are also made within text in lines 79-87 and Table 1.

Response 5: We thank the reviewer for the suggestion, and we have clarified this in the figure legend of Figure 2, as well as section 2.2, lines 129-135 in the revised manuscript.

Response 6: We thank the reviewer for the suggestion. The splenocytes were ex vivo re-stimulated with Met e 1, and culture supernatants were collected 72h post-culture for ELISA to measure cytokine levels. This is now included in the method section (section 4.5, lines 468-472) of the revised manuscript .

Response 7: Blood samples were collected 24 h after each challenge (i.e. first and second challenges) to compare the allergen-specific IgE level before and after vaccine treatment. This is now included in the method section in line 465.

Response 8: We thank the reviewer for this valuable comment. We also noticed that pMEM49 and pMED171 could induce different subsets of Treg cells and have briefly discussed this in lines 301-305. However, we have focused on examining the role of the classical Foxp3+Treg cells in the present study with depletion and adoptive transfer experiments considering that this is the most widely studied Treg subset in AIT and with well-established method, therefore allowing us to compare our pMED171 treatment data with other similar studies in the literature. Therefore, we did not study the other subsets, including surface expression of LAP in this study. More detailed analyses on characterizing the phenotype and function of the different Treg subsets induced by pMEM49 and pMED171 are projected in follow up studies. We also thank the reviewer’s suggestion of the co-administration of pMEM49 and pMED171 in future experiments. We have addressed these points in the revised manuscript at Line 312-315.

Response 9: We thank the reviewer for this comment. The pMED171-induced CD25+Treg cells for transfer have been characterized by stimulating them with Met e 1 and assessed for cytokine levels. IL-10 level was undetectable, and we therefore believe that only one set of Treg cell, the CD4+CD25+Foxp3+Treg cells, were transferred that generate the results described in the manuscript. The details are described at lines 218-221 in the revised manuscript.

Response 10: We thank the reviewer for the comment. Literature has suggested that the gut homing signals on T cells were imprinted by intestinal CD103+ dendritic cells by retinoic acid (RA) (Iwata et al. 2004 Immunity 21:527-38; Johansson-Lindbom et al. 2005 J Exp Med 202: 1063-73). We therefore performed qPCR to characterize the dendritic cells in the MLN and also small intestine with a specific emphasis on the RA signaling pathway (e.g. Aldh1a1-3 and Wnt signaling pathways). However, expression of these genes in the pMED171 group did not differ from the control groups. We are therefore considering other possible pathways. With the lack of sufficient supportive data, we did not include these analyses in the present study.

Reviewer 2 Report

Dear Authors, 

I have reviewed your article entitled "Modulating Shrimp Tropomyosin-Mediated Allergy: Hypoallergen DNA Vaccines Induce Regulatory T Cells to Reduce Hypersensitivity in Mouse Model".

The article builds on work by the same group, is well written and experiments well designed. Conclusions are mostly supported by the data presented and the work is a valuable contribution to the field.

Nevertheless, the article presents very little on the mechanistic associated to the generation of the different Treg type responses. Also, the lack of impact on IgG responses as reported in previous work is puzzling and suggests relevant impacts of the delivery method.

Major issues:

1-Considering the AIT stated limitations, the study does present potential advances on the efficacy of such a treatment, and the Treg induction shown (with pMED171) is indicative of good possibility of achieving a relevant degree of long-term protection. However, the 7 days interval between treatment and re-challenge is too short to convincingly show that the effect is durable. A longer interval would be required in order to add more weight to  that message (>30 days). This is relevant when considering clinical potential.

2-The Treg role in the effect seen with the pMEM49 is debatable. The increase shown (Fig3B and 3C) is probably dependent on CD25, and CD25 levels on foxp3neg cells also seem somewhat increased in the dot-plot shown. IL2 was not evaluated in these experiments, but an increase in IL2 levels might result in the differences shown. Are total Foxp3pos % changed in pMEM49? As all further experiments are performed with pMED171, it may still occur that effects of pMEM49 are not Treg mediated. Have Treg depletion and Treg transfer experiments been performed in the context of pMEM49 treatment? The data shown is not sufficient to claim that effects pMEM49 treatment are indeed due to Treg generation, although may be near enough to suggest the possibility. Discussion should make this distinction between the two tested treatments clearer, as they are (lines 282-286), statements are not supported by the data. 

Minor points, comments:

1- Line 80, eliminate "that" after "while".

2- Figure 3E, has similar data been evaluated on other sites? Are the increased Splenic Tregs poised to migrate to the gut via higher expression of gut homing or chemokine receptors? Note that transfer experiments are performed with cells taken from the spleen..

3-Figure 5C,E: Please align plots, both CD25 and Foxp3 intensities (MFIs?) seem higher in pMED171 plots, but it is not easy to evaluate this with not aligned plots. 

4-Line 284: Please Correct "independent"

5- (Comment) : Why do these treatments induce migration to the gut? 

Best regards 

Round 2

Reviewer 1 Report

Most of my comments have been answered in the new version of the manuscript and I believe that the modified version of this manuscript can be consider for its publication.